# On-Ground Calibration of a Nano-G Accelerometer for Micro-Vibration Monitoring in Space on a Dual-Axis Indexing Device

**DOI:** 10.3390/s25206289

**Published:** 2025-10-10

**Authors:** Yang Zhou, Zhi Li, Qiangwei Xu, Xiangchun Li

**Affiliations:** 1School of Information Science and Technology, Wenhua College, Wuhan 430074, China; yang_zhou@hust.edu.cn; 2School of Physics, Huazhong University of Science and Technology, Wuhan 430074, China; phdlizhi@hust.edu.cn; 3College of Electrical Engineering, Henan University of Technology, Zhengzhou 450001, China; xqw@haut.edu.cn

**Keywords:** nano-*g* accelerometer, parameter identification, two-axis indexing device, multi-position calibration, on-orbit experiment

## Abstract

High-sensitivity accelerometers are essential for spacecraft micro-vibration monitoring. This study proposes a procedure to facilitate precise on-ground calibration of such accelerometers with a limited operational range by rotating to multiple positions with its input axis mounted along the horizontal tilt axis of a two-axis indexing device. Each single-axis accelerometer unit of a self-developed tri-axial nano-*g* accelerometer was respectively tested with its various reference axes along the rotation axis for identifying the parameters of their model equations including higher-order terms. The minute tilt axis deviation of the test equipment from the horizontal plane and the accelerometer’s higher-order response to gravity during calibration are corrected for application in the microgravity environment. Errors of accelerometer biases and scale factors are satisfactorily improved, respectively, to ±2% and ±0.01 m*g*, by at least one order of magnitude. Parameters of all three units of the accelerometer are unified into one coordinate frame defined by the accelerometer mounting surface. Acceleration measured by our accelerometer shows consistency with the other collocated one in a space mission.

## 1. Introduction

High-sensitivity accelerometers with self-noise below 1 μ*g*/√Hz (*g* ≈ 9.8 m/s^2^) are widely used in ground precision gravity measurement [1,2], inertial navigation, and spacecraft micro-vibration monitoring [3,4]. Monitoring ground motions or gravity anomalies provides valuable information about mass distribution and seismic activity [5]. Civil applications include resource exploration, earthquake prediction, and geophysical surveys, while military uses include underground bunker detection and gravity-assisted navigation [6,7]. Accelerometers with nano-*g* resolution or better are also sought for satellite gravity gradiometry [8,9], terrestrial or space-based gravitational wave detection [10,11,12], and planetary exploration missions [13,14].

Parameterized accelerometer models describe the mathematical relationship between input and output, forming the basis for interpreting accelerometer measurements. Accurate calibration is essential to determine model parameters and compensate for errors. In inertial navigation, accelerometers in an IMU experience multi-directional inputs, including constant and time-varying accelerations [15]. Nonlinearities can introduce variations in the DC component, known as vibration rectification error (VRE), which accumulates over time and necessitates nonlinearity calibration [16,17]. In aerospace applications, microgravity environments need to be monitored for potential acceleration fluctuations caused by radiation pressure, gas leaks, air drag, etc. [18]. The model parameters of an accelerometer in space might be considerably different from those calibrated on-ground due to the different gravitational fields. This situation applies to monitoring seismic activities on Mars or the Moon using accelerometers, too. Ground-based calibration can provide applicable parameters for spaceborne accelerometers through careful error correction, avoiding the complexity of on-orbit calibration [19]. In the realm of natural resource exploration, the airborne gravity gradiometer (AGG) comprises four sub-ng precision accelerometers [20]. Precisely calibrated accelerometers would greatly facilitate the instrumentation [21]. The aforementioned examples underscore the importance of precisely calibrating these accelerometers on the ground.

Conventional calibration methods, such as centrifuge tests [22], linear vibration tests, or gravity field tumbling tests [23,24,25,26,27], face challenges in the use of high-precision, small-range accelerometers. For example, the nano-*g* tri-axial accelerometer tested in this paper has an input range of ±2 mg, restricting tilt angles to a few mrad on the ground. Moreover, test equipment noise, environmental disturbances, and accelerometer bias drift limit the achievable signal-to-noise ratio and degrade calibration precision. Techniques such as rotational modulation on tilting turntables or square-wave/double-frequency signals have been proposed to identify specific higher-order parameters [26,27,28,29,30,31,32,33], but they are limited to certain error effects of specific sensors such as micro-seismometers and electrostatic suspension accelerometers [34,35,36,37,38]. A multi-position tumble test in the gravity field allows rapid identification of full accelerometer models, yet requires careful procedure design and error analysis when applied to nano-*g* accelerometers. Particularly, validating model parameters for a gravitational field different from the calibration remains critical.

This paper focuses on the precise calibration of a tri-axial accelerometer for microgravity applications. We first establish an error model for multi-position testing and provide a comprehensive input–output expression including second-order effects for a single-axis unit rotating on a dual-axis indexing device. Calibration procedures separate mutually coupled parameters by rotating the unit along different reference axes, allowing independent evaluation for each axis. Installation and positioning errors are compensated, and parameters are unified into a common coordinate frame based on the reference defined by the precision fixture and dual-axis device. Finally, a self-developed tri-axial nano-*g* accelerometer (MEMS-M^3^) is tested, and measurement uncertainties of each parameter are evaluated. The method is validated by comparing on-orbit acceleration measurements from MEMS-M^3^ with those from other collocated accelerometers.

## 2. Accelerometer and Model Equation

### 2.1. Description of the Accelerometer to Be Tested

In this paper, theoretical models and experiments are based on a tri-axial MEMS accelerometer, called MEMS-M^3^, with a self-noise of 2–5 n*g*/√Hz within 0.1–10 Hz and an input range of more than ±2 m*g*, as shown in Figure 1. It was developed for measuring the micro-vibration of a spacecraft. As shown in Figure 2, the MEMS-M^3^ consists of three orthogonal, single-axis accelerometer units. The accelerometer housing has a smooth datum plane and bottom plane for aligning the accelerometer with the mounting platform on the spacecraft. The MEMS-M^3^ has a volume of 105 × 90 × 115 mm^3^, a weight of 1.2 kg, and a power consumption of 3 W [39].

The input range of the MEMS-M^3^ is approximately ±2 m*g*. During calibration in the ground gravity field, it is not possible for all three unit of the MEMS-M^3^ to work simultaneously. Therefore, each of the three unit were tested separately.

### 2.2. Model Equation of Accelerometer

To describe the input–output relationship of a single-axis accelerometer unit [23], we utilize the mathematical model as follows:(1)$$\begin{array}{c} \;\;E=K_{1}(K_{0}+a_{i}+K_{2}a_{i}^{2}+K_{\mathrm{oo}}a_{o}^{2}+K_{\mathrm{pp}}a_{p}^{2}+K_{\mathrm{ip}}a_{i}a_{p}\\ +K_{\mathrm{io}}a_{i}a_{o}+K_{\mathrm{op}}a_{o}a_{p}+\delta_{o}a_{p}-\delta_{p}a_{o}+\varepsilon ) \end{array}$$
where E is the accelerometer output (V); $K_{1}$ is the scale factor (V/*g*); $K_{0}$ is the bias (g); $a_{i}$, $a_{o}$, $a_{p}$ are the applied acceleration components (*g*) along the input axis (IA), output axis (OA), and pendulous axis (PA) of the accelerometer, respectively; $K_{2}$, $K_{\mathrm{oo}}$, $K_{\mathrm{pp}}$ are the nonlinearity coefficients (*g*/*g*^2^); $K_{\mathrm{ip}}$, $K_{\mathrm{io}}$, $K_{\mathrm{op}}$ are the cross-coupling coefficients (*g*/*g*^2^); $\delta_{o}$ and $\delta_{p}$ are the misalignment (rad) of the IA with respect to the input reference axis about the OA and the PA, respectively; and ε is the remnant noise (*g*).

## 3. Calibration Method of High-Precision Tri-Axial Accelerometer

### 3.1. Test Setup and Coordinate Systems

As shown in Figure 3, the tri-axial accelerometer is mounted on the top surface of a dual-axis turntable (used as an indexing device) through the fixture. The turntable is used to orient the accelerometer, and it can respectively rotate within ±180° around the tilt axis ($O_{2}y_{2}$) and the rotation axis ($O_{2}z_{2}$). Rotation around the tilt axis ($O_{2}y_{2}$) provides a deviation of the turntable mounting surface from the horizontal plane, while the turntable mounting surface can rotate around the rotation axis ($O_{2}z_{2}$) at a certain deviation angle. The variations generated by the turntable are within ±1″ (*k* = 1 in this paper) when actuated in a closed-loop mode. The perpendicularity error between the rotation and tilt axis is within ±3″ (*k* = 1 in this paper). Ideally, the initial zero positions of the tilt axis nominally defined by the turntable and the $O_{2}x_{2}$ axis are supposed to be overlapped within the horizontal plane.

Based on the spatial relationship between the accelerometer and the turntable rotation axis, the accelerometer can be mounted in two configurations. Figure 4 illustrates the PA rotation mounting position with the PA along the turntable rotation axis, and the OA rotation mounting position with the OA along the turntable rotation axis.

Several intermediate frames have to be introduced in order to practically determine the attitudes of all three sensitive axes in a unified frame. Our calibration process involves six coordinate systems, as shown in Figure 5.

The laboratory coordinate system is $O_{1}x_{1}y_{1}z_{1}$, where the $x_{1}O_{1}y_{1}$-plane is horizontal and the $O_{1}z_{1}$ axis is along the direction of gravity.The turntable coordinate system is $O_{2}x_{2}y_{2}z_{2}$, where the $O_{2}x_{2}$ axis is overlapped with the tilt axis of the turntable and the $O_{2}y_{2}$ is orthogonal to the $O_{2}x_{2}$ within the rotation surface when the tilt angle is set to the nominal zero position. Therefore, the misalignment between the coordinate systems $O_{2}x_{2}y_{2}z_{2}$
and $O_{1}x_{1}y_{1}z_{1}$ is fixed and mainly caused by the installation error of the turntable.The rotating coordinate system is $O_{3}x_{3}y_{3}z_{3}$, which is fixed to the mounting surface of the turntable and moves together with the accelerometer as the turntable rotates. Figure 6 shows how the turntable coordinate system $O_{2}x_{2}y_{2}z_{2}$ is transformed to the rotating coordinate system $O_{3}x_{3}y_{3}z_{3}$.The fixture coordinate system is $O_{4}x_{4}y_{4}z_{4}$, which is realized by rotating the $O_{3}x_{3}y_{3}z_{3}$ system about the $O_{3}z_{3}$ axis with an angle ξ, and where ξ is the mounting error of the fixture relative to the turntable platform, as shown in Figure 7. The accelerometer can be mounted with different configurations through the fixture.The datum plane coordinate system is $O_{5}x_{5}y_{5}z_{5}$, where the axes are defined by the accelerometer housing the datum plane and bottom surface, as shown in Figure 2. The attitudes of every sensitive axis separately measured in the fixture coordinate system $O_{4}x_{4}y_{4}z_{4}$ have to be unified in the datum plane coordinate system $O_{5}x_{5}y_{5}z_{5}$.The accelerometer coordinate system $O_{6}x_{6}y_{6}z_{6}$ is determined by the IAs of the three units of the tri-axial accelerometer. The difference between $O_{6}x_{6}y_{6}z_{6}$ and $O_{5}x_{5}y_{5}z_{5}$ is the accelerometer misalignment of its three sensitive axes.

### 3.2. Traditional Multi-Position Calibration Method

The traditional multi-position calibration method involves tilting the turntable at a specified angle and performing a multi-position test around the rotation axis, as shown in Figure 6. This makes it possible to partly obtain the coefficients of the accelerometer model equation from a single rotation experiment. To identify the complete parameters such as the bias, the misalignment, and second-order nonlinear coefficients, multiple sets of multi-position rotation tests are performed at different tilt angles (e.g., 0°, 45°, and 90°) [23,26,27].

This method is primarily designed for accelerometers with input ranges no less than ±1 *g*. It is sufficiently precise to consider only the turntable and rotation coordinate systems in many cases, and many minor errors can be neglected. The transformation matrix from the turntable coordinate system to the rotation coordinate system is(2)$$C_{2}^{3}=\left[\begin{array}{ccc} \cos\theta & -\sin\theta & 0\\ \sin\theta & \cos\theta & 0\\ 0 & 0 & 1 \end{array}\right]\left[\begin{array}{ccc} \cos\gamma & 0 & \sin\gamma \\ 0 & 1 & 0\\ -\sin\gamma & 0 & \cos\gamma \end{array}\right]$$
where γ is the tilt angle of the turntable, and θ is the rotation angle of the turntable.

The gravitational acceleration is represented by the vector $g={\left(0,0,g\right)}^{T}$. Taking the OA rotation mounting position as an example, the gravitational acceleration components projected on the accelerometers PA, IA, and OA are(3)$$\left[\begin{array}{c} a_{p}\\ a_{i}\\ a_{o} \end{array}\right]=C_{2}^{3}g=\left[\begin{array}{c} g\sin\gamma \cos\theta \\ g\sin\gamma \sin\theta \\ g\cos\gamma \end{array}\right]$$

In a simplified case, a linear model equation is generally adopted to describe the accelerometer’s response including terms related to the bias, the scale factor, and the misalignment as follows:(4)$$\begin{array}{l} E=K_{1}\left(K_{0}+a_{i}+\delta_{o}a_{p}+\delta_{p}a_{o}\right)\\ =K_{1}K_{0}+K_{1}g\sin\gamma \sin\theta +\delta_{o}g\sin\gamma \cos\theta -\delta_{p}g\cos\gamma \end{array}$$

Equation (4) is sufficiently precise for measuring micro-vibrations of a spacecraft in a microgravity environment. However, we cannot readily obtain the parameters in (4) by simply calibrating on the ground with the same equation due to the significant difference in the ambient gravity field. A more complete model should be adopted for calibration and then used to predict its on-orbit behavior. The corresponding equation is written as(5)$$\begin{array}{cc} E & =K_{1}[K_{0}+\delta_{o}g\sin\gamma \cos\theta -\delta_{p}g\cos\gamma \\ & +0.5\left(K_{\mathrm{pp}}-K_{2}\right)g^{2}\sin^{2}\gamma +K_{\mathrm{oo}}g^{2}\cos^{2}\gamma \\ & +\left(1+K_{\mathrm{io}}g\cos\gamma \right)g\sin\gamma \sin\theta +0.5K_{\mathrm{op}}g^{2}\sin2\gamma \cos\theta \\ & +0.5K_{\mathrm{ip}}g^{2}\sin^{2}\gamma \sin2\theta +0.5\left(K_{\mathrm{pp}}-K_{2}\right)g^{2}\sin^{2}\gamma \cos2\theta ] \end{array}$$

It can be seen that a single rotation experiment at a tilt angle *γ* is not sufficient to separate all the parameters in Equation (5). For example, $K_{\mathrm{io}}g\cos\gamma$ is dependent in exactly the same way as the scale factor. The lumped effect of these two parameters leads to a mixed term $\left(1+K_{\mathrm{io}}g\cos\gamma \right)g\sin\gamma \sin\theta$. The bias $K_{0}$ of the accelerometer is also inseparable from $-\delta_{p}g\cos\gamma +0.5\left(K_{\mathrm{pp}}-K_{2}\right)g^{2}\sin^{2}\gamma +K_{\mathrm{oo}}g^{2}\cos^{2}\gamma$ at a fixed tilt angle. The same problem arises at some other harmonic components of θ. To determine all these parameters, multiple sets of multi-position rotation tests are performed at different tilt angles (e.g., γ = 0°, 45°, 90°), utilizing their dependency on the tilt angle. It is important to separate them, especially when accelerometers under test are intended for use in a different attitude on the ground or in an environment with a different gravity field [23]. Otherwise, some error effects during calibration on the ground would be mistaken as coefficients in a simplified form like Equation (4).

However, technical challenges occur when using the aforementioned method to calibrate a high-precision accelerometer with a limited input range. Taking MEMS-M^3^ as an example, the tilt angle of the MEMS-M^3^ IA relative to the horizontal plane cannot exceed 5 mrad on ground. In this case, it is found not suitable to simply follow the conventional test procedure of multi-position rotations by calibrating at a series of small tilt angles (such as 0, 2.5, 5 mrad) because the gravity component along the rotation axis (that is, the acceleration along the OA or PA) negligibly changes no more than (1 − cos 0.005) = 25 ppm. In addition, for small tilt angle tests, the relative errors resulting from installation error, angular positioning error, and environmental variations become significant. Detailed analysis reveals that certain errors would be further amplified by hundreds of times at small angles.

### 3.3. Improved Scheme Based on Separately Rotating Along Each of the Three Reference Axes of a Single-Axis Accelerometer Unit

It is essential to characterize a high-precision accelerometer using a complete nonlinear accelerometer model equation since its nonlinearity coefficients are typically large. This complete characterization would allow us to more precisely assess the performance of accelerometers in an application scenario different from the calibration environment. Here, we employ all six coordinate systems described in section A to build a calibration model for a single-axis accelerometer unit [34,35].

First, we start with analyzing the turntable installation error for a single-axis accelerometer unit. As shown in Figure 8, the system $O_{1}x_{1}y_{1}z_{1}$ is rotated around the $O_{1}x_{1}$ axis by an angle $\beta_{1}$ and around the $O_{1}y_{1}$ axis by an angle $\beta_{2}$ to obtain the turntable coordinate system $O_{2}x_{2}y_{2}z_{2}$. The rotation around the $O_{1}z_{1}$ axis has no influence on the test. The transformation matrix from the $O_{1}x_{1}y_{1}z_{1}$ to the $O_{2}x_{2}y_{2}z_{2}$ can be written as(6)$$\begin{array}{cc} C_{1}^{2} & =\left[\begin{array}{ccc} 1 & 0 & 0\\ 0 & \cos\beta_{1} & \sin\beta_{1}\\ 0 & -\sin\beta_{1} & \cos\beta_{1} \end{array}\right]\;\left[\begin{array}{ccc} \cos\beta_{2} & 0 & \sin\beta_{2}\\ 0 & 1 & 0\\ -\sin\beta_{2} & 0 & \cos\beta_{2} \end{array}\right]\\ & \approx \left[\begin{array}{ccc} 1 & 0 & \beta_{2}\\ 0 & 1 & \beta_{1}\\ -\beta_{2} & -\beta_{1} & 1 \end{array}\right]. \end{array}$$

Due to the fixture mounting error and accelerometer misalignment, the accelerometer coordinate system $O_{6}x_{6}y_{6}z_{6}$ does not precisely align with the rotating coordinate system $O_{3}x_{3}y_{3}z_{3}$. Taking the OA rotation mounting position as an example, Figure 9 illustrates that the angle between the IA and the $O_{3}y_{3}$ axis can be decomposed into the out-of-plane installation error angle $\delta_{1}$ and the in-plane installation error angle $\delta_{2}$. $\delta_{1}$ is the angle between the IA and the $x_{3}O_{3}y_{3}$ plane, and $\delta_{2}$ is the angle between the $O_{3}y_{3}$ axis and the projection of the IA in the $x_{3}O_{3}y_{3}$ plane. The coordinate systems $O_{4}x_{4}y_{4}z_{4}$ and the $O_{5}x_{5}y_{5}z_{5}$ are introduced to describe the accelerometer misalignment, which will be discussed in the next section.

In the system, the unit vector along the IA is(7)$$n_{I}={\left[-\cos\delta_{1}\sin\delta_{2},\cos\delta_{1}\cos\delta_{2},\sin\delta_{1}\right]}^{T}$$

According to the orthogonality relations $n_{I}\cdot \;n_{P}=0$ and $n_{I}\times n_{P}=n_{O}$, we obtain the transformation matrix from $O_{6}x_{6}y_{6}z_{6}$ to $O_{3}x_{3}y_{3}z_{3}$, which is(8)$$C_{6}^{3}=\left(n_{P},n_{I},n_{O}\right)\approx \left[\begin{array}{ccc} 1 & -\delta_{2} & 0\\ \delta_{2} & 1 & \delta_{1}\\ 0 & \delta_{1} & -1 \end{array}\right]$$

Similarly, the transformation matrix $C_{3}^{6}$ at the PA rotation mounting position is(9)$$C_{3}^{6}=\left[\begin{array}{ccc} 1 & \delta_{2} & 0\\ -\delta_{2} & 1 & \delta_{1}\\ 0 & -\delta_{1} & 1 \end{array}\right]$$

After obtaining the error matrices $C_{1}^{2}$ and $C_{3}^{6}$, we can calculate the accurate accelerometer output expression. When the accelerometer is mounted in the OA rotation mounting position, the gravitational acceleration components projected on the accelerometer PA, IA, and OA are(10)$$\left[\begin{array}{c} a_{p}\\ a_{i}\\ a_{o} \end{array}\right]=C_{3}^{6}C_{2}^{3}C_{1}^{2}g\approx g\left[\begin{array}{c} \left(\gamma +\beta_{2}\right)\cos\left(\theta -\delta_{2}+\beta_{1}/\gamma \right)\\ \left(\gamma +\beta_{2}\right)\sin\left(\theta -\delta_{2}+\beta_{1}/\gamma \right)-\delta_{1}\\ -1+\gamma \beta_{2} \end{array}\right]$$

By substituting (10) into (1), we obtain the accelerometer output, as represented in (11). Expressed as the sum of individual harmonic components of the angular position, the accelerometer output is(11)$$\begin{array}{cc} E\approx & K_{1}[K_{0}-g\delta_{1}+K_{2}g^{2}{\delta_{1}}^{2}+K_{\mathrm{oo}}g^{2}+K_{\mathrm{io}}g\delta_{1}\\ & \;-K_{\mathrm{ip}}g^{2}\delta_{1}\varphi \cos\theta +\left(1-2K_{2}g\delta_{1}-K_{\mathrm{io}}g\right)g\varphi \sin\theta \\ & \;+0.5K_{\mathrm{ip}}g^{2}\varphi^{2}\sin2\theta -0.5K_{2}g^{2}\varphi^{2}\cos2\theta ]\\ & \approx K_{1}^{*}[K_{0}^{*}+g\varphi \sin\theta +\left(\delta_{2}-\beta_{1}/\gamma \right)g\varphi \cos\theta \\ & \;+0.5K_{\mathrm{ip}}g^{2}\varphi^{2}\sin2\theta -0.5K_{2}g^{2}\varphi^{2}\cos2\theta ], \end{array}$$
where(12)$$\begin{array}{cc} \varphi & =\gamma +\beta_{2},\\ K_{0}^{*} & =K_{0}-g\delta_{1}+K_{\mathrm{io}}g\delta_{1}+K_{\mathrm{oo}}g^{2}\approx K_{0}-g\delta_{1},\\ K_{1}^{*} & =K_{1}\left(1-2K_{2}g\delta_{1}-K_{\mathrm{io}}g\right)\approx K_{1}\left(1-K_{\mathrm{io}}g\right). \end{array}$$

Note that Equation (11) is a simplified version of the accelerometer output expression. Many of the terms in the expression are omitted when their influences on the accelerometer output are estimated to be approximately 1 μ*g* or less. The estimations are based on typical values of coefficients listed in Table 1. These values are obtained from the accuracy of the test equipment, the machining errors of the mechanical parts, and the test results of other methods [28,31].

When the accelerometer is installed in the PA rotation mounting position, similar calculation derives(13)$$\left[\begin{array}{c} a_{o}\\ a_{i}\\ a_{p} \end{array}\right]\approx g\left[\begin{array}{c} \left(\gamma +\beta_{2}\right)\cos\left(\theta -\delta_{2}+\beta_{1}/\gamma \right)\\ \left(\gamma +\beta_{2}\right)\sin\left(\theta -\delta_{2}+\beta_{1}/\gamma \right)+\delta_{1}\\ 1-\gamma \beta_{2} \end{array}\right]$$

Substituting (19) into (1), we obtain(14)$$\begin{array}{cc} E & \approx K_{1}[K_{0}+\delta_{1}+K_{2}g^{2}{\delta_{1}}^{2}+K_{\mathrm{pp}}g^{2}+K_{\mathrm{ip}}g\delta_{1}\\ & \;+K_{\mathrm{io}}g^{2}\delta_{1}\varphi \cos\theta +\left(1-2K_{2}g\delta_{1}+K_{\mathrm{ip}}g\right)g\varphi \sin\theta \\ & \;+0.5K_{\mathrm{io}}g^{2}\varphi^{2}\sin2\theta -0.5K_{2}g^{2}\varphi^{2}\cos2\theta ]\\ & \approx K_{1}^{*}[K_{0}^{*}+g\varphi \sin\theta +\left(\delta_{2}-\beta_{1}/\gamma \right)g\varphi \cos\theta \\ & \;+0.5K_{\mathrm{io}}g^{2}\varphi^{2}\sin2\theta -0.5K_{2}g^{2}\varphi^{2}\cos2\theta ], \end{array}$$
where(15)$$\begin{array}{cc} \varphi & =\gamma +\beta_{2},\\ K_{0}^{*} & =K_{0}+g\delta_{1}+K_{\mathrm{ip}}g\delta_{1}+K_{\mathrm{pp}}g^{2}\approx K_{0}+g\delta_{1},\\ K_{1}^{*} & =K_{1}\left(1-2K_{2}g\delta_{1}+K_{\mathrm{ip}}g\right)\approx K_{1}\left(1+K_{\mathrm{ip}}g\right). \end{array}$$

Note that $\delta_{1}$ and $\delta_{2}$ in (11) and (14) are not equal due to differences in the mounting position.

We can see from (11) or (14) that $K_{0}^{*}$ differs from the bias $K_{0}$ by an additional value of $g\delta_{1}$ and $K_{1}^{*}$ differs from the scale factor $K_{1}$ by an relative increase of $K_{\mathrm{io}}g$ or $K_{\mathrm{ip}}g$. For a high-precision accelerometer, these differences could be significant. To illustrate, if the $K_{1}^{*}$ obtained from the ground calibration is applied directly as the scale factor to measurement in a space microgravity environment, the relative error generated by the cross-coupling term can reach ±10% on the order of magnitude with a cross-coupling coefficient of ±0.1 *g*/*g*^2^. This error needs to be corrected for routine micro-seismic monitoring. Furthermore, there is an error $\delta_{r}=\beta_{1}/\gamma$ mixed with $\delta_{2}$. Provided that γ is in the order of a few mrad, the error of $\delta_{r}$ would increase by several hundred times. ±1 arcsec of $\beta_{1}$ can lead to a $\delta_{2}$ error of approximately ±1 mrad. A slight zero-position deviation of the tilt angle from the horizontal would lead to a large error of the misalignment $\delta_{2}$, which is very different from the situation that the tilt angle is comparable to π/2 rad or so. This imposes a stringent requirement on the accuracy of the turntable.

Based on the aforementioned analyses, when calibrating a high precision accelerometer with a limited operating range, we have developed a viable calibration scheme of rotating the single-axis accelerometer along each of its three reference axes.

At first, we conduct the test both in the OA rotation mounting position and/or in the PA rotating position. For example, after we separate $K_{\mathrm{io}}$ using the PA rotation test result, we can subtract $K_{\mathrm{io}}g$ from $K_{1}^{*}$ of the OA rotation test to obtain $K_{1}$ according to (12).

Secondly, we utilize an electronic gradienter to measure the deviation angle of the turntable tilt axis from the horizontal. The uncertainty of angular measurement is 0.05 arcsec, much better than the angular precision of the turntable. The test procedure involves fixing the gradienter on the turntable, aligning its sensitive axis parallel to the turntable tilt axis $O_{2}x_{2}$, and setting the tilt angle to the nominal zero position of the turntable. Subsequently, we conduct the multi-position rotation test. Due to the tilt axis deviation of the turntable from the horizontal, the gradienter output β will exhibit periodic changes with respect to the rotation angle θ. The two deviation angles $\beta_{1}$ and $\beta_{2}$ can be derived according to the following function relationship:(16)$$\beta =\beta_{1}\cos\theta +\beta_{2}\sin\theta$$

Thirdly, we determine the misalignment $\delta_{1}$ and the cross-axis nonlinearity coefficients using a novel IA rotating method. The procedure involves mounting the accelerometer with its IA approximately parallel to the turntable tilt axis, and changing the tilt angle to make the accelerometer IA rotate around the tilt axis. In this case, the gravitational acceleration components on the OA, IA, PA, and the accelerometer output are shown respectively in (17) and (18) as(17)$$\left[\begin{array}{c} a_{o}\\ a_{i}\\ a_{p} \end{array}\right]=g\left[\begin{array}{c} \sin\gamma \\ \delta_{1}\cos\gamma -\delta_{2}\sin\gamma +\beta_{1}\\ \cos\gamma \end{array}\right]$$(18)$$\begin{array}{cc} E/K_{1} & =K_{0}+g\beta_{1}+\delta_{1}g\cos\gamma -\delta_{2}g\sin\gamma +\\ & 0.5g^{2}[K_{2}\delta^{2}+K_{\mathrm{oo}}+K_{\mathrm{pp}}+K_{\mathrm{ip}}\delta_{1}-K_{\mathrm{io}}\delta_{2}+\\ & (K_{\mathrm{pp}}-K_{\mathrm{oo}}+K_{2}\delta^{2}+K_{\mathrm{ip}}\delta_{1}+K_{\mathrm{io}}\delta_{2})\cos2\gamma \\ & +(K_{\mathrm{op}}-2K_{2}\delta_{1}\delta_{2}+K_{\mathrm{io}}\delta_{1}-K_{\mathrm{ip}}\delta_{2})\sin2\gamma ]\\ & \approx K_{0}^{*}+\delta_{1}g\cos\gamma -\delta_{2}g\sin\gamma +\\ & 0.5g^{2}(K_{\mathrm{pp}}-K_{\mathrm{oo}})\cos2\gamma +0.5g^{2}K_{\mathrm{op}}\sin2\gamma , \end{array}$$
where(19)$$\begin{array}{l} \delta^{2}=\delta_{1}^{2}+\delta_{2}^{2},\\ K_{0}^{*}=K_{0}+g\beta_{1}. \end{array}$$

According to Equation (18), we can determine $\delta_{1}$ and $\delta_{2}$ using the cos*γ* and sin*γ* components, and $K_{\mathrm{pp}}-K_{\mathrm{oo}}$ and $K_{\mathrm{op}}$ using the cos2*γ* and sin2*γ* components. After substituting the measured $\delta_{1}$ into (11) or (14), $K_{0}$ is determined. It should be noted that $\delta_{2}$ can be separately obtained using (11), (14), or (18). The advantage of using (18) to obtain $\delta_{2}$ is that the signal of the related term has a magnitude of $\delta_{2}g$, which is much larger than the magnitude $\delta_{2}g\varphi$ in (11) and (14). Therefore, the $\delta_{2}$ obtained in the IA rotating test should be more accurate. So far, all the coefficients in the simplified Equation (4) can be derived with our proposed procedure.

### 3.4. Unified Misalignment Calibration of the Three Accelerometer Units

Misalignment between the sensitive axes of the tri-axial accelerometer and its housing is a crucial parameter in applications such as spacecraft micro-vibration monitoring. In our space experiments, the accelerometer’s datum plane coordinate system is aligned with the spacecraft coordinate system [37]. Therefore, it is necessary to measure misalignment between the sensitive axis of each accelerometer unit and the spacecraft coordinate system, which are represented by the angular errors $\delta_{o}$ and $\delta_{p}$ in(20)$$C_{5}^{6}=\left[\begin{array}{ccc} 1 & \delta_{p} & 0\\ -\delta_{p} & 1 & \delta_{o}\\ 0 & -\delta_{o} & 1 \end{array}\right]$$

In the previous section, we solely addressed the measurement of $\delta_{1}$ and $\delta_{2}$, which correspond to the matrix $C_{3}^{6}$. This matrix encompasses two factors: $C_{5}^{6}$ and $C_{4}^{5}$. Thus, by measuring $C_{3}^{4}$ and $C_{4}^{5}$, we can obtain $C_{5}^{6}$ and the misalignment as follows:(21)$$\begin{array}{l} C_{3}^{6}=C_{5}^{6}C_{4}^{5}C_{3}^{4},\\ C_{5}^{6}=C_{3}^{6}{\left(C_{3}^{4}\right)}^{-1}{\left(C_{4}^{5}\right)}^{-1}. \end{array}$$

$C_{3}^{4}$ comprises one unknown parameter, namely the fixture mounting error ξ. The value of ξ can be determined by measuring the deviation of the fixture datum plane from the horizontal with an electronic gradienter when the tilt angle is set to 90°. The angular accuracy of the turntable ensures that this angular error can be measured with an uncertainty of 2 arcsecond. We can also measure the mounting error ξ using a gradienter with its in-plane misalignment calibrated.

$C_{4}^{5}$ is measured directly with an uncertainty of 2 arcsecond by the CMM (Coordinate Measuring Machine). Assuming that the system $O_{4}x_{4}y_{4}z_{4}$ is rotated by an angle of $\alpha_{x}$ around the $O_{4}x_{4}$ axis, next rotated by an angle of $\alpha_{y}$ angle around the $O_{4}y_{4}$ axis, and finally rotated by an angle of $\alpha_{z}$ around the $O_{4}z_{4}$ axis, $O_{5}x_{5}y_{5}z_{5}$ is obtained. The approximate expressions of $C_{3}^{4}$ and $C_{4}^{5}$ are(22)$$C_{3}^{4}\approx \left[\begin{array}{ccc} 1 & -\xi & 0\\ \xi & 1 & 0\\ 0 & 0 & 1 \end{array}\right],\;C_{4}^{5}\approx \left[\begin{array}{ccc} 1 & \alpha_{z} & -\alpha_{y}\\ -\alpha_{z} & 1 & \alpha_{x}\\ \alpha_{y} & -\alpha_{x} & 1 \end{array}\right]$$

### 3.5. Data Processing Procedures

To separate the nonlinearity coefficients and scale factors, multi-position rotation tests need be conducted separately in various rotation mounting positions. Figure 10 presents the data processing flowchart of the multi-position calibration experiment. With the outputs of the turntable and sensors (accelerometer or electronic gradienter) at multiple rotation positions in each mounting position, the least squares method is used to fit the harmonic coefficients X. The parameters of the sensors under test are then estimated according to their relationship with the harmonic coefficients, dependent on the chosen model equations. The misalignment of every accelerometer unit in its own rotating coordinate system $O_{3}x_{3}y_{3}z_{3}$ is unified into the datum plane coordinate system by coordinate transformation. The flowchart of the accelerometer misalignment calibration is shown in Figure 11. Additional information is obtained from the measurement using a CMM and a gradienter.

## 4. Experiments and Discussion

This section provides an overview of the experimental procedure, taking the Z axis of the MEMS-M^3^ as an illustrative example.

### 4.1. Testing at Input Range of ±2 mg

To obtain the parameters $K_{0}^{*}$ and $K_{1}^{*}$ in Equation (14), a multi-position rotation test was conducted with an accelerometer input range of ±2 m*g*. The MEMS-M^3^ is fixed to the turntable with the X-axis accelerometer unit in the PA rotation mounting position.

In order to improve the measurement accuracy, the dwell time at each position was optimized through Allan deviation analysis. A typical output data segment is chosen to calculate the Allan deviation when MEMS-M^3^ is in a static state. The acceleration deviation is shown in Figure 12 as a function of the time interval. The dwell time is chosen to be the interval where the lowest Allan deviation is achieved. The lowest deviation occurs within 5–8 s for all three units and does not increase much within tens of seconds. The dwell time at each position was finally chosen to be 8 s, considering that the Allan deviation of the Y-unit increases significantly below the interval of 8 s, and the overall test duration is a couple of minutes.

During the rotation test, we found that the accelerometer output could easily reach its limit at specific positions, even when the tilt angle was sufficiently small. This over-range phenomenon can be attributed to the $K_{0}^{*}$ defined in Equation (14). $K_{0}^{*}$ consists of the real bias of the accelerometer and the constant gravity component along the input axis due to misalignment out of the rotation plane. A relatively large value of $K_{0}^{*}$ leads to a smaller range along one direction of the input axis. Therefore, we only collected data from positions where the accelerometer outputs are within the output range. Figure 13 shows the output of one accelerometer unit as a function of the rotation angle.

Fitting the data according to (20), we have(23)$$\begin{array}{cc} E & =K_{1}^{*}\left[K_{0}^{*}+\varphi \sin\theta +\left(\delta_{2}-\beta_{1}/\gamma \right)\varphi \cos\theta \right]\\ & =2096\times \left[-0.004581+0.321^{\circ }\sin\left(\theta +0.0177\right)\right]. \end{array}$$

Hence, $K_{0}^{*}$ is −4.581 ± 0.001 m*g* and $K_{1}^{*}$ is 2096 ± 2 V/*g*. Note that the nonlinearity terms are omitted here since they are at second harmonics of the rotation angle θ and the magnitude is insignificant at a small tilt angle γ. In order to obtain the real $K_{0}$ and $K_{1}$, we need to further consider the tilt angle error of the turntable and the gravity coupling through the accelerometer misalignment and the nonlinearity.

### 4.2. Calibration of Turntable Installation Error

The turntable tilt error was measured using an electronic gradienter, as shown in Figure 14. Fitting according to (16) gives $\beta_{1}=10.27\pm 0.05\prime \prime$ and $\beta_{2}=3.29\pm 0.05\prime \prime$. Figure 15 depicts the scale factors measured from the multi-position experiments at different tilt angles. The scale factor, in the green color, is derived by directly fitting the data, and the relative uncertainty, in the blue color, increases significantly as the tilt angle decreases. The corrected scale factor, in the red color, after compensating for the turntable tilt error is constant within 0.1% at different tilt angles. This proves that it is advantageous to correct the tilt error even when calibrating the scale factor.

### 4.3. Testing at Input Range of ±13 mg

We temporarily expanded the accelerometer measurement range to ±13 m*g* in order to complete the multi-position rotation test covering the range of 0–360° at a relatively large tilt angle. The expanded range is beneficial for more precisely calibrating the accelerometer misalignment and nonlinearity coefficient, which is considered to be independent of the measurement range. As shown in Figure 16, during the multi-position PA rotation test, MEMS-M^3^ works normally at all positions. The multi-position PA rotation test is used to identify $K_{2}$ and $K_{\mathrm{io}}$. A multi-position IA rotation test was performed to identify $\delta_{1}$, $\delta_{2}$, $K_{\mathrm{pp}}-K_{\mathrm{oo}}$, and $K_{\mathrm{op}}$. The analysis results of the multi-position PA rotation test and IA rotation test are summarized in Table 2 [36]. All the measured values are given with their associated standard uncertainties. The uncertainties presented in the table are obtained from the fitting calculations and are estimated based on the statistical distribution assumption of the residuals, which assumes a normal (Gaussian) distribution. Substituting the $\delta_{1}$ into Equation (23), we obtain $K_{0}=-2.27\pm 0.01\;mg$. Similarly, experiments were performed in the OA rotation mounting position to determine *K*_ip_. Substituting the *K*_ip_ into Equation (15), we obtain $K_{1}=2048\left(8\right)\;V/g$.

### 4.4. Misalignment Test

As shown in Figure 17, we measured the tilt axis error of the turntable by fitting with the output of an electronic gradienter at a nominal zero position of the turntable. Next, the spatial relationship between $O_{4}x_{4}y_{4}z_{4}$ and $O_{5}x_{5}y_{5}z_{5}$ was measured using a CMM, which has a positioning accuracy of 1 μm. According to Equation (21), the misalignment relative to the coordinate system defined by the accelerometer housing the datum plane and the bottom surface was derived and is summarized in Table 3.

### 4.5. Results and Uncertainty Analysis

The test results for the three axes of MEMS-M^3^ are shown in Table 4.

Table 5 lists all measurement uncertainties evaluated with typical values in our test using the new procedure. The relative uncertainty of the scale factor is 2%, which is limited by the uncertainties of $K_{\mathrm{ip}}$ or $K_{\mathrm{io}}$ (0.02 *g*/*g*^2^), although the relative uncertainty of $K_{1}^{*}$ is 0.1%. Nevertheless, it represents a dramatic error decrease by one order of magnitude, compared to the deviation of 20% before correction, for application in the microgravity environment. Similarly, the error of accelerometer biases is reduced from 1 mg to 0.01 mg by an order of magnitude. With all these coefficients, we are able to describe the response of the accelerometer with better accuracy in different application environments.

### 4.6. On-Orbit Data Analysis

This section processes the MEMS-M^3^ on-orbit data using the ground-calibrated parameters. The above test results determined the installation angles of the three axes of the three reference axes of the accelerometer relative to the smooth datum plane and the bottom plane of the accelerometer housing, as shown in Figure 2. When installed in the spacecraft frame, aligning the accelerometer mounting surface to the prepared reference plane of the spacecraft allows the angles on the spacecraft to be known from ground-calibrated misalignment. MEMS-M^3^ in the test spacecraft was launched by the Long March 5B rocket (CZ-5B) into near-Earth orbit at 10:00 on 5 May 2020, and returned to the ground at 5:00 on May 8, 2020. More information about the MEMS-M^3^ can be found in [39]. During this period, MEMS-M^3^ acquired data for about 11 h. From the data recorded by the MEMS-M^3^, it is evident that the X and Y channels are working properly on-orbit. Figure 18 depicts the acceleration measured on-orbit using the X channel of MEMS-M^3^ and a commercial STIM300 IMU, with the DC component subtracted to align the data. The STIM300 has an input range of ±5 *g*, but with an accuracy of only 0.1 m*g*. The MEMS-M^3^ has a limited input range (about ±2.3 m*g*), causing occasional over-range in its data. As can be seen from the normal data segment, the MEMS-M^3^ has a lower noise level than the STIM300, meaning that it can distinguish smaller vibrations. Figure 18 gives a comparison of the two accelerometer measurements over a time interval. Both sensors observed a large acceleration step at 27,000 s, which was most likely caused by the separation between the core and cargo modules of the spacecraft. This step provided a useful calibration signal to verify the scale factor of the MEMS-M^3^. We performed median filtering on the data from both sensors with a 500 s moving window, in order to accurately observe the step signal. The acceleration step measured by the STIM300 is 0.185 ± 0.025 m*g*. While the corresponding step in the output voltage of the MEMS-M^3^ is 0.375 ± 0.054 V, the magnitude is calculated to be 0.177 ± 0.025 m*g* according to its scale factor calibrated on the ground. The two results are marginally consistent within the 2*σ* interval. This preliminary on-orbit result manifests the accuracy of calibrating the scale factor on ground by compensation to meet the needs of space applications.

## 5. Conclusions

This paper presents a method for calibrating the scale factor, bias, misalignment, and nonlinearity coefficients of high-precision accelerometers with limited measurement range using an on-ground dual-axis indexing device. We propose the identification of these accelerometer model parameters using multi-position tests, respectively rotating around the accelerometer OA, PA, and IA. A custom n*g*-precision tri-axial accelerometer MEMS-M^3^ for monitoring micro-vibrations in space is employed to verify the coefficients of the accelerometer model equation calibrated on-ground. Compared with conventional test methods such as multi-point rotation on an indexing device tilted at different angles [23,24,25], parameter deviations caused by installation errors, cross-coupling effects, etc., are estimated and are effectively compensated for. The error of accelerometer biases is reduced from ±20% to ±2%, and that of the scale factors is reduced from 1 m*g* to 0.01 m*g* on the order of magnitude, for application in the microgravity environment. The on-orbit acceleration magnitude measured by our MEMS-M^3^ shows agreement with other collocated accelerometers with a lower precision and higher measurement range. This research presents a feasible test procedure and data processing method for on-ground parameter calibration of high-precision accelerometers for an application environment with different gravity. Further verification of more model parameters is desired in the future. These findings can be of significance for further exploring the role of high-precision accelerometers in the fields of terrestrial precision gravity measurement and aerospace gravity gradient measurement.

## Figures and Tables

**Figure 1 sensors-25-06289-f001:**
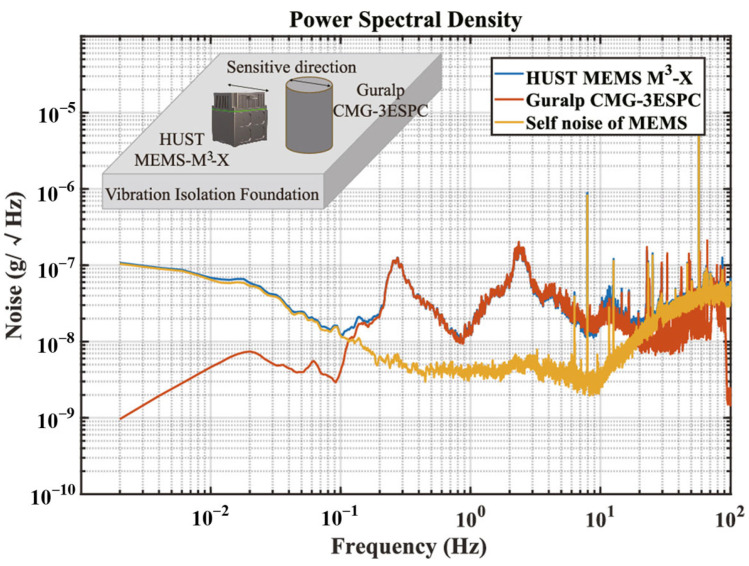
Noise floor test results of MEMS-M^3^ X-axis [30].

**Figure 2 sensors-25-06289-f002:**
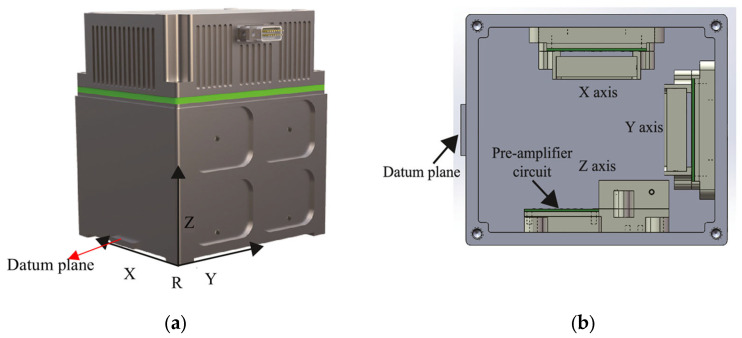
(**a**) The tri-axial accelerometer MEMS-M^3^; (**b**) internal structure of the MEMS-M^3^.

**Figure 3 sensors-25-06289-f003:**
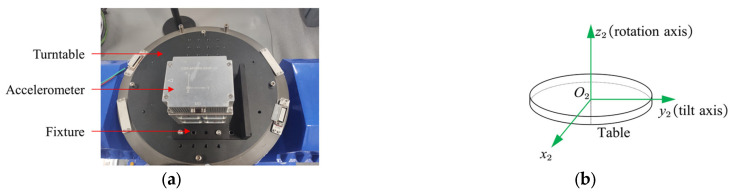
Calibration configuration: (**a**) is the test setup and (**b**) is the orthogonal turntable coordinate system.

**Figure 4 sensors-25-06289-f004:**
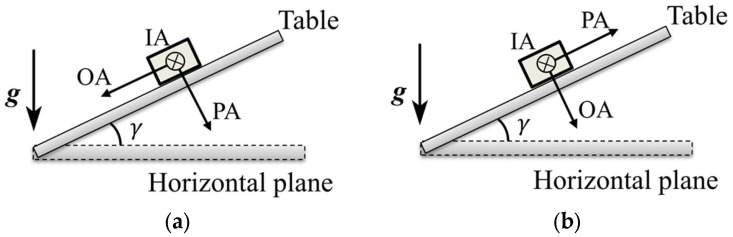
Schematic mounting configurations of the accelerometer on the turntable: (**a**) represents the PA rotation mounting position, and (**b**) is the OA rotation mounting position.

**Figure 5 sensors-25-06289-f005:**
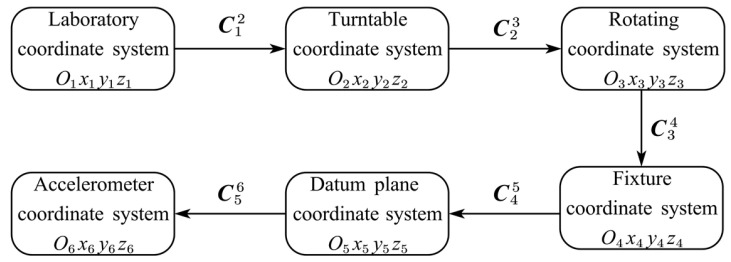
Definition of coordinate systems.

**Figure 6 sensors-25-06289-f006:**
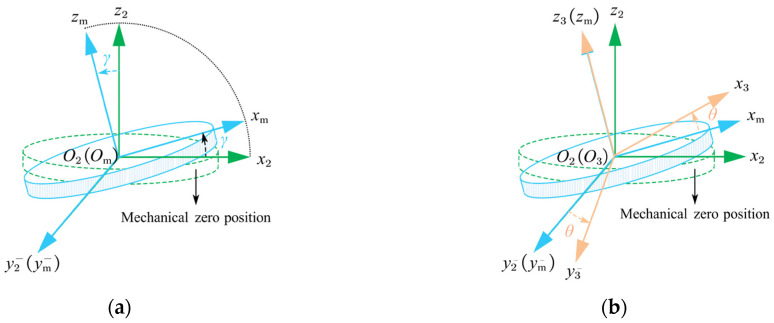
Schematic diagram of the turntable and the rotation coordinate systems. The $O_{m}x_{m}y_{m}z_{m}$ is an intermediate coordinate system in order to illustrate the two-step rotation operation. (**a**) $O_{2}x_{2}y_{2}z_{2}$ rotates around the tilt axis ($O_{2}y_{2}$) and becomes $O_{m}x_{m}y_{m}z_{m}$. (**b**) $O_{m}x_{m}y_{m}z_{m}$ rotates around the rotation axis ($O_{m}z_{m}$) and becomes $O_{3}x_{3}y_{3}z_{3}$.

**Figure 7 sensors-25-06289-f007:**
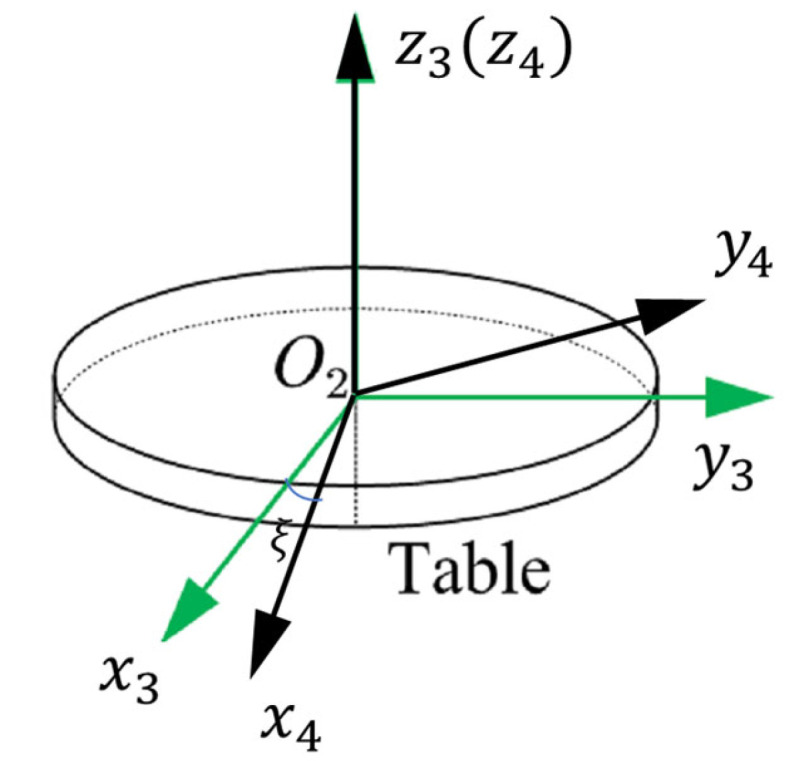
Schematic diagram of the fixture coordinate system.

**Figure 8 sensors-25-06289-f008:**
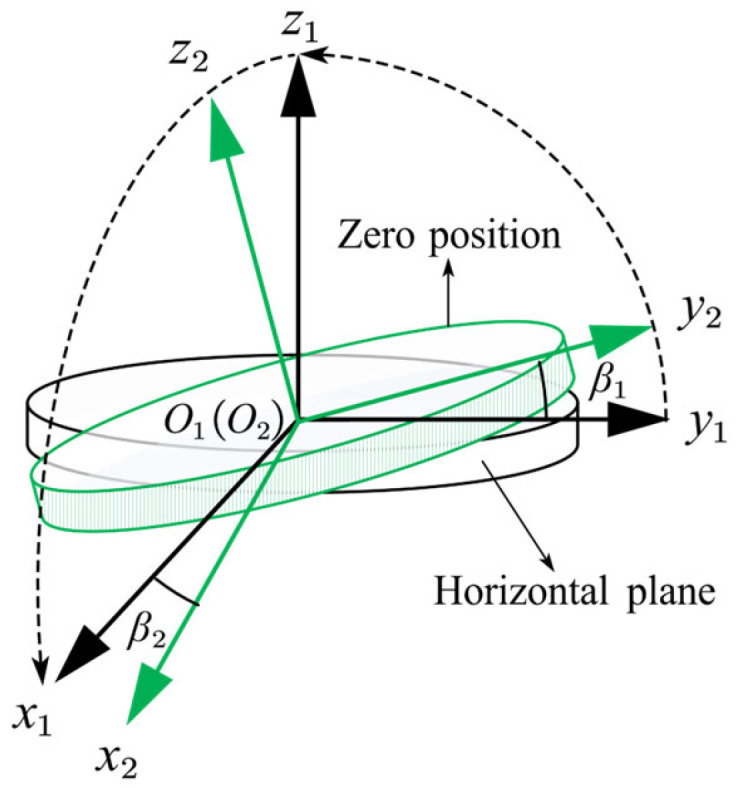
Schematic diagram of the turntable installation error.

**Figure 9 sensors-25-06289-f009:**
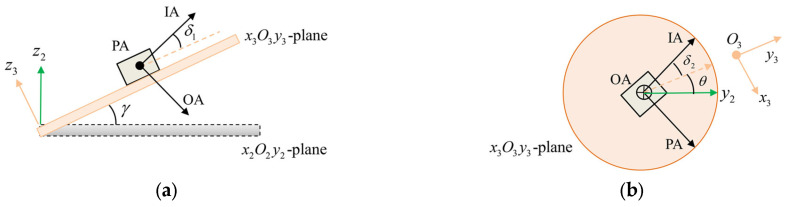
Schematic diagram of the error between the accelerometer and the rotating coordinate system: (**a**) side view; (**b**) top view.

**Figure 10 sensors-25-06289-f010:**
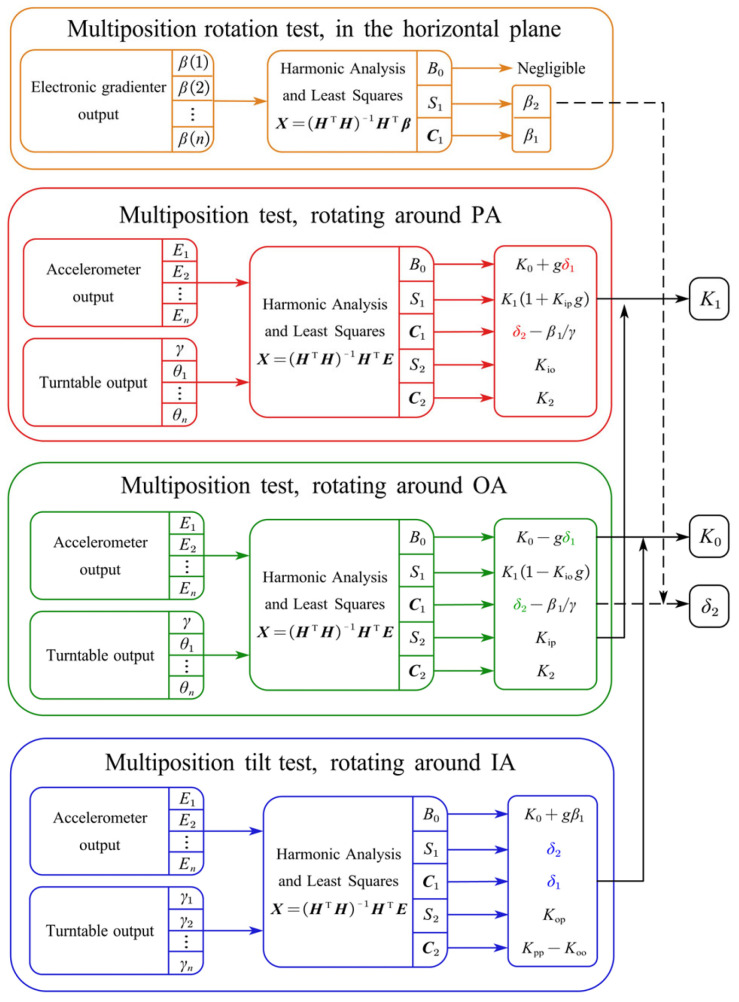
Data processing flowchart of multi-position tests.

**Figure 11 sensors-25-06289-f011:**
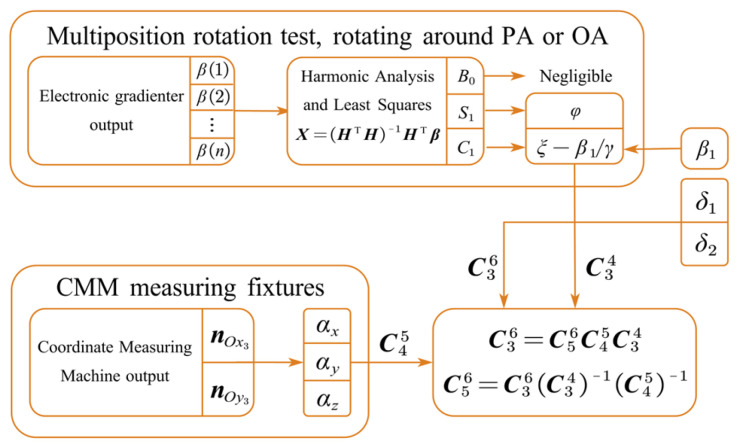
Data processing flowchart of calibrating accelerometer misalignment.

**Figure 12 sensors-25-06289-f012:**
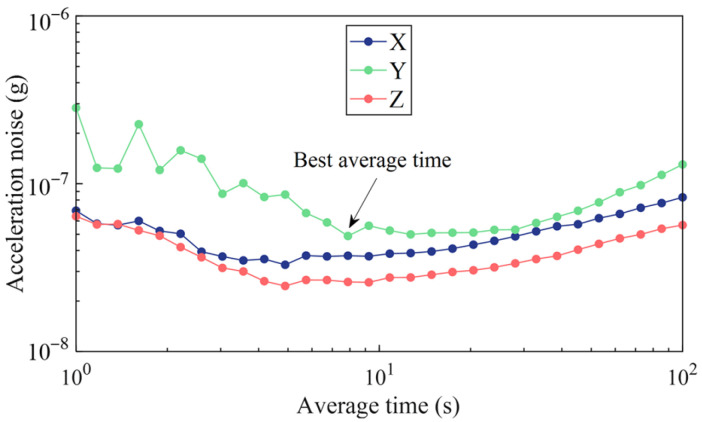
Allan deviation of MEMS-M^3^ output in a static state.

**Figure 13 sensors-25-06289-f013:**
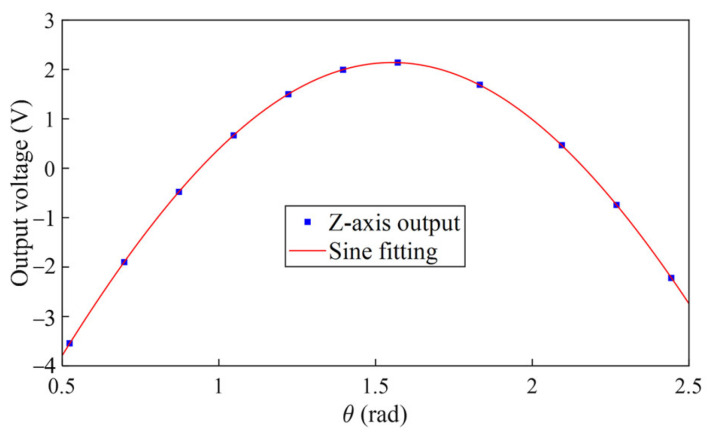
Multi-position rotation test data at ±2 mg input range, where *γ* = 0.32°.

**Figure 14 sensors-25-06289-f014:**
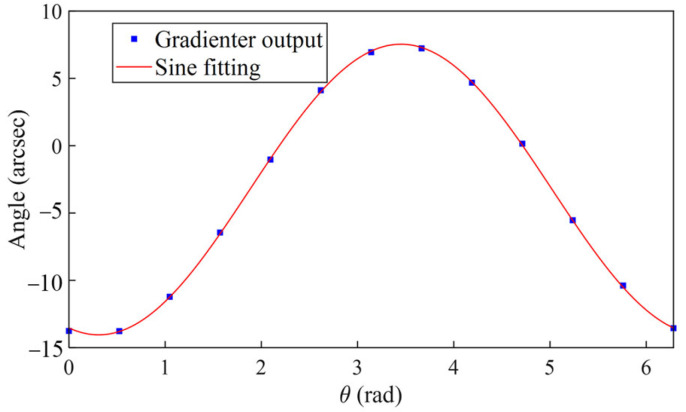
Test result of the gradienter when the turntable tilt angle is set to a nominal zero position.

**Figure 15 sensors-25-06289-f015:**
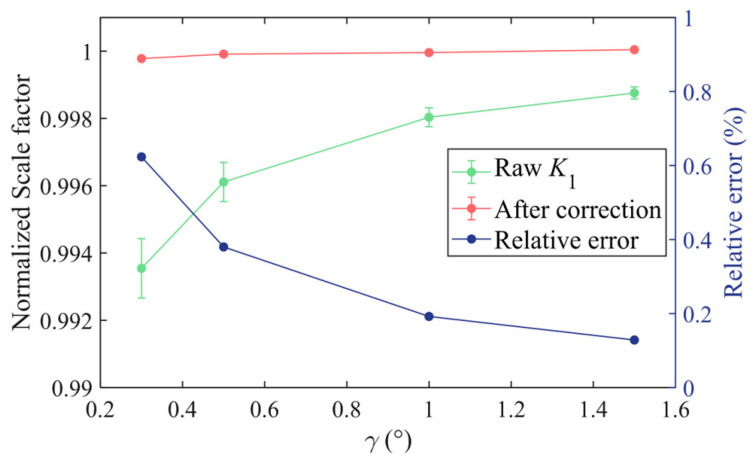
Scale factors measured at different tilt angles.

**Figure 16 sensors-25-06289-f016:**
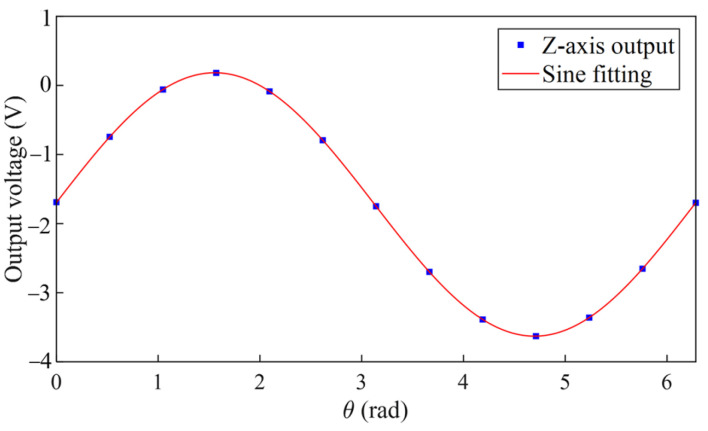
Accelerometer output in a multi-position PA rotation test with a tilt angle γ=0.32°.

**Figure 17 sensors-25-06289-f017:**
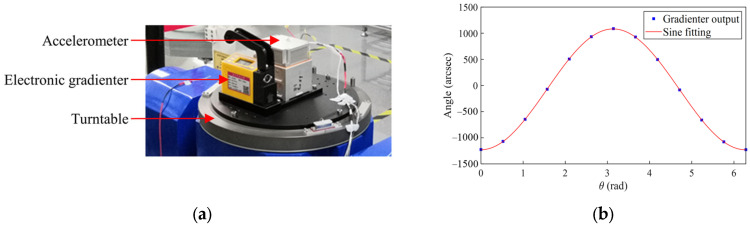
Measuring fixture installation error. (**a**) Photo of the test setup; (**b**) gradienter output as a function of the rotation angle.

**Figure 18 sensors-25-06289-f018:**
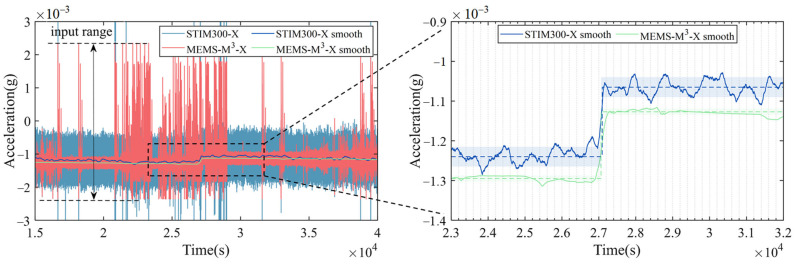
On-orbit X-axis acceleration measured by MEMS-M3 and STIM300.

**Table 1 sensors-25-06289-t001:** Parameter estimates.

Parameters	Meaning	Typical Values
$\delta_{1}$, $\delta_{2}$	Installation error of IA	±5 mrad
φ γ	Tilt angle of the turntable	±5 mrad
$K_{0}$	Bias of accelerometers	±1 m*g*
$K_{2}$, $K_{\mathrm{ip}}$, $K_{\mathrm{io}}$	Second-order term coefficients related to IA	±0.1 *g*/*g*^2^
$K_{\mathrm{oo}}$, $K_{\mathrm{pp}}$, $K_{\mathrm{op}}$	Second-order term coefficients unrelated to IA	±0.1 m*g*/*g*^2^

**Table 2 sensors-25-06289-t002:** Test results of the Z axis.

	Parameters	Results
Multi-position PA rotation	$K_{2}$	0.22 ± 0.02 *g*/*g*^2^
	$K_{\mathrm{io}}$	0.15 ± 0.03 *g*/*g*^2^
Multi-position IA tilt	$\delta_{1}$	−2.31 ± 0.01 mrad
	$\delta_{2}$	5.98 ± 0.01 mrad
	$K_{\mathrm{pp}}-K_{\mathrm{oo}}$	0.12 ± 0.01 m*g*/*g*^2^
	$K_{\mathrm{op}}$	0.11 ± 0.02 m*g*/*g*^2^

**Table 3 sensors-25-06289-t003:** Test results of misalignment.

Transformation Matrix	Parameters	Values (mrad)
$C_{3}^{6}$	$\delta_{1}$	−2.31 ± 0.01
	$\delta_{2}$	5.98 ± 0.01
$C_{3}^{4}$	ξ	−3.1 ± 0.3
$C_{4}^{5}$	$\alpha_{x}$	−1.1 ± 0.1
	$\alpha_{y}$	1.6 ± 0.1
	$\alpha_{z}$	6.7 ± 0.1
$C_{5}^{6}=C_{3}^{6}{\left(C_{3}^{4}\right)}^{-1}{\left(C_{4}^{5}\right)}^{-1}$	$\delta_{o}$	−6.6 ± 0.3
	$\delta_{p}$	5.2 ± 0.3

**Table 4 sensors-25-06289-t004:** Test results of three axes of MEMS-M^3^.

Parameters	X Axis	Y Axis	Z Axis
$K_{1}$ (V/*g*)	2119 ± 6	2029 ± 6	2048 ± 8
$K_{0}$ (m*g*)	−2.49 ± 0.01	−4.66 ± 0.01	−2.27 ± 0.01
$K_{2}$ (*g*/*g*^2^)	0.23 ± 0.02	0.27 ± 0.02	0.22 ± 0.02
$K_{\mathrm{io}}$ (*g*/*g*^2^)	0.29 ± 0.02	0.15 ± 0.03	0.15 ± 0.03
$K_{\mathrm{ip}}$ (*g*/*g*^2^)	0.012 ± 0.003	0.009 ± 0.003	0.023 ± 0.003
$K_{\mathrm{op}}$ (m*g*/*g*^2^)	0.043 ± 0.004	0.108 ± 0.006	0.11 ± 0.02
$K_{\mathrm{pp}}-K_{\mathrm{oo}}$ (m*g*/*g*^2^)	0.288 ± 0.008	0.093 ± 0.003	0.12 ± 0.01
$\delta_{o}$ (mrad)	1.0 ± 0.3	0.9 ± 0.3	−6.6 ± 0.3
$\delta_{p}$ (mrad)	6.5 ± 0.3	3.3 ± 0.3	5.2 ± 0.3

**Table 5 sensors-25-06289-t005:** Evaluation of measurement uncertainty.

Parameters	Error Sources	Values	Synthetic Uncertainties
$K_{1}^{*}$	Uncertainty of $\beta_{2}$	0.05″	0.1%
	Accelerometer drift	10 μ*g*	
$K_{1}$	Uncertainty of $K_{\mathrm{ip}}$ or $K_{\mathrm{io}}$	0.02 *g*/*g*^2^	2%
$K_{0}$	Accelerometer drift	10 μ*g*	0.01 m*g*
	Uncertainty of $\delta_{1}$	0.01 mrad	
$K_{2}$,$K_{\mathrm{ip}}$,$K_{\mathrm{io}}$	Accelerometer drift	2 μ*g*	0.02 *g*/*g*^2^
$K_{\mathrm{op}}$,$K_{\mathrm{pp}}-K_{\mathrm{oo}}$	Fitting error	13 μ*g*	0.01 m*g*/*g*^2^
$\delta_{1}$,$\delta_{2}$	Fitting error	13 μ*g*	0.01 mrad
$\delta_{o}$,$\delta_{p}$	Small-angle approximation	50 ppm	0.3 mrad
	Uncertainty of $\delta_{1}$, $\delta_{2}$	0.01 mrad	
	Accuracy of CMM	1 μm	
	Uncertainty of ξ	0.3 mrad	
ξ	Uncertainty of $\beta_{1}$	0.05″	0.3 mrad
	Electronic gradienter drift	1″	

## Data Availability

Data available on request due to legal restrictions.

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
