# Peer review of "On-Ground Calibration of a Nano-G Accelerometer for Micro-Vibration Monitoring in Space on a Dual-Axis Indexing Device"

_sensors, 2025, doi:10.3390/s25206289_

Round 1
Reviewer 1 Report
Comments and Suggestions for Authors
A. General information
The content of the manuscript is original and interesting, however the paper should be more concise; it should not resemble a report. At the present form, it is too long, especially the introduction; maybe it would be possible e.g., to delete the whole section 3.2? Some crucial problems have not been addressed.
Use of English is generally correct, nevertheless I have proposed few improvements.
Presentation of the content should be improved with regard to the following issues of a general character:
A1. the first 2 sentences of the abstract fit rather the introduction,
A2. the abstract is the most strategic part of the paper; thus, it should be a concise summary of the most important findings and results; at the present form, it resembles rather a detailed presentation of the whole manuscript instead of communicating the most essential content - a high-level view of the research; it contains no information about the results, it lacks numerical values included, note that usually only a few of the first sentences of the abstract are displayed while using a web browser – so they should be the most representative in order to increase chances for future citations,
A3. Conclusions contain no information about the results, with the numerical values included,
A4. the authors failed to clearly state/emphasize in Conclusions what new was found out, what problems had been discovered thanks to the undertaken study, and what is very important: how their findings could be applied in a more general manner,
A5. the authors failed to compare in Conclusions their findings with results of similar studies by other researchers,
B. The most crucial errors
Here are the most important errors, that have to be addressed:
B1. Too little is revealed about the tested accelerometers; in my opinion there is no reason to refer to the accelerometers as “nano-g”, since their noise is in the order of mg (1000 times higher!); what are the basic operational parameters of the accelerometers (measurement range, bandwidth, dimensions of the package, etc.)? Can you provide any related publication?
B2. Such crucial issues related to MEMS accelerometers as: long-term and temperature drifts, amplitude and phase attenuation over frequency are totally disregarded in the text,
B3. In the case of generating such small variations of the acceleration for the purpose of calibration, too little is said about the used turntable: what is the axial play associated with the rotation about the vertical axis?, what is the error of perpendicularity between rotation and tilt axis of the turntable? What about the influence of variations generated by the turntable while actuated? In line 255 we read: “the error of the dividing device is relatively negligible” – could you evaluate value of the error?.
B4. The authors specified in Fig. 4 quite a complicated structure of the test rig. However, once the sensor had been calibrated, how to determine its misalignment while installed in the spacecraft frame?
C. Detailed remarks:
C1. Delete the first paragraph of the introduction
C2. I propose to delete the whole fragment: “Firstly… not corrected” - line 530-536
C3. are all the digits of the numerical values presented (e.g. in Tab. 5-7) statistically significant? what about a proper rounding corresponding to relevant errors?
C4. Line 47: „involve nuclear submarine” – what do you mean specifically?
C5. Fig. 1 is not clear: what are the dimensions, how are the accelerometers fixed to the housing?; the characters are too small,
C6. Line 187 – the zeta angle has not been illustrated in any figure
C7. A suggestion: it would be more convenient to convert the “output volage” in figure 6, 14, 15 into acceleration
C8. Table 1 and 6 provide values of the uncertainty; however we do not know what is the probability distribution (normal?); did the authors statistically test a respective hypothesis?
C9. “Table 2. Measurement uncertainty at small tilt angles” – what do you mean by small tilt angles; what is their magnitude?
C10. Line 281: „Along The Three Axis of A Single-Axis Accelerometer Unit” – first „three axes”; second three axes of a single-axis accelerometer sounds illogical
C11. Figure 7: for sure, order of beta 1 and beta 2 does matter; which was applied first? beta 1?
C12. Line 433: „The tri-axial accelerometer” – tri-axial?; „with one unit of the three” – what three units?
C13. Line 505: „From the data recorded by the MEMS M3, it is evident that the X and Y channels are working properly on orbit, but the Z channel is abnormal” – no related figure was presented
D. References:
D1. there are no references published recently in the journal of Sensors (as to prove relevance of the submitted manuscript to the journal; a recent reference should be dated most preferably 2023-2024 to build the upcoming journal Impact Factor and CiteScore); otherwise, one may get an impression that the manuscript better fits some other journals, which are cited more frequently
D2. most of the references (27) are quoted only in the Introduction,
D3. most of the cited items are by authors, who originate from Asia, so references of a more worldwide origin would be more convincing
D4. 3 items cited in the main body are rather not a satisfactory number; add some references related to the essential content of the manuscript
Comments on the Quality of English LanguageSome lingual errors
E1. I propose to change "home-made" to "custom" – line 538
E2. I propose to change “ranges more than” to “ranges greater than” – line 78
E3. Line 104: “Gravity field tumbling” - find a better wording
E4. Line 109: “There are few existing literatures involved” - find a better wording
E5. I propose to change “challenges occurs” to “challenges occur” - line 260
E6. I propose to change “is as small as” to „is in the order of” - line 341
E7. I propose to change “10% on the order of magnitude” to „10% of the order of magnitude” - line 339
E8. I propose to change “Based on those above-mentioned analyses of the increased error when calibrating” to „Based on the aforementioned analyses when calibrating” - line 347
E9. I propose to change “make the accelerometer IA rotating” to „make the accelerometer IA rotate” - line 365
E10. I propose to change “Up to now” to „so far” - line 375
E11. Figure 17: “smooth” – do you mean filtered?
Reviewer 2 Report
Comments and Suggestions for Authors
This paper presents a comprehensive calibration methodology based on a dual-axis indexing device for a self-developed tri-axial nano-g level MEMS accelerometer (MEMS-M³). The proposed approach covers model establishment, error analysis, experimental design, and data processing, forming a complete and practically valuable calibration procedure with significant engineering applicability.
- Equation (5) establishes a relatively complete model that accounts for second-order nonlinearities. Given the nano-g level precision, have the effects of higher-order nonlinear terms been considered, as they may introduce non-negligible errors?
- Sections 3.2 and 3.3 describe the traditional calibration method and the improved proposed method, respectively. It is recommended to condense these sections and use a clearer format—such as a comparative table—to summarize the differences between the two methods, facilitating readers' understanding. Additionally, the order of experiments in Section 4 should be aligned with the calibration procedure outlined in Section 3.3.
- The on-orbit validation section mentions abnormal data from the Z-axis but provides no further explanation. It would be beneficial to either include an explanation or supplement with normal data segments.
- The Introduction on the first page contains template instructions that have not been replaced. The section is also relatively lengthy and could be made more concise. Some sentences throughout the manuscript are long and complex; it is advised that the authors carefully review and improve the clarity and fluency of the English expression.
